# GoT-R1: Unleashing Reasoning Capability of Autoregressive Visual Generation with Reinforcement Learning

**Chengqi Duan**[1*]   **Rongyao Fang**[2*]   **Yuqing Wang**[1*]   **Kun Wang**[3]   **Linjiang Huang**[4]

**Xingyu Zeng**[3,5]   **Hongsheng Li**[2]   **Xihui Liu**[1†]

[1]HKU MMLAB   [2]CUHK MMLAB   [3]Sensetime   [4]Beihang University   [5]SUAT
`duancq24@connect.hku.hk, xihuiliu@eee.hku.hk`

## Abstract

Visual generation models have made remarkable progress in creating realistic images from text prompts, yet struggle with complex prompts that specify multiple objects with precise spatial relationships and attributes. Effective handling of such prompts requires explicit reasoning about the semantic content and spatial layout. We present GoT-R1, a framework that applies reinforcement learning to enhance semantic-spatial reasoning in autoregressive visual generation models. Leveraging the natural affinity between autoregressive architectures and sequential reasoning, our approach builds upon the Generation Chain-of-Thought framework to enable models to autonomously discover effective reasoning strategies beyond predefined templates. To achieve this, we propose a dual-stage multi-dimensional reward framework that leverages MLLMs to evaluate both the reasoning process and final output, enabling effective supervision across the entire generation pipeline. The reward system assesses semantic alignment, spatial accuracy, and visual quality in a unified approach. Experimental results demonstrate significant improvements on T2I-CompBench and GenEval benchmark, particularly in compositional tasks involving precise spatial relationships and attribute binding. GoT-R1 advances the state-of-the-art in autoregressive image generation by successfully transferring sophisticated reasoning capabilities from language models to the visual generation domain. Code is available at `https://github.com/gogoduan/GoT-R1`.

## 1 Introduction

Visual generation (Podell et al., 2023; Ramesh et al., 2022; Saharia et al., 2022; Esser et al., 2024; Nichol et al., 2021; Labs, 2024; Rombach et al., 2022) has witnessed great advances in recent years, enabling the creation of diverse and realistic visuals from natural language descriptions. Despite their impressive capabilities, these models often struggle with complex and compositional prompts that specify multiple objects with precise spatial relationships and attributes (Huang et al., 2025). This limitation stems from their direct mapping from text embeddings to visual features without explicit reasoning of the compositional structure of the desired scene. The Generation Chain-of-Thought (GoT) (Fang et al., 2025) framework tackles this challenge by introducing an intermediate semantic-spatial reasoning process that decomposes complex prompts into explicit object descriptions with location coordinates before image generation, significantly improving compositional fidelity. While GoT was originally designed for diffusion models, autoregressive architectures offer natural advantages for sequential reasoning tasks due to their step-by-step token generation process. Nevertheless, GoT's reasoning capability—regardless of architecture—is constrained by supervised fine-tuning with human-defined templates, limiting the model's ability to autonomously discover more effective reasoning strategies. As shown in Fig. 1, we observe that GoT-generated reasoning chains can be unfaithful to the prompt despite following templates well.

---

[*]Equal Contribution
[†]Corresponding Author

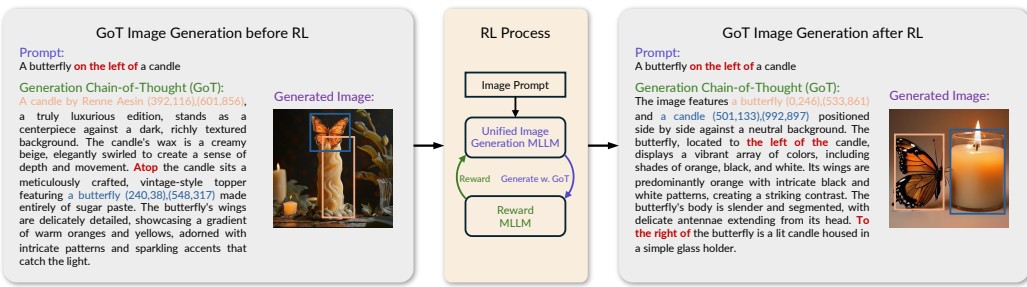

Figure 1: GoT-R1 enhances visual generation through reinforcement learning. This figure demonstrates the improvement from a GoT-finetuned model (**left**) to the RL-trained GoT-R1 model (**right**). The model before RL generates spatially misaligned reasoning process. The RL process enhances the model's semantic-spatial reasoning capabilities, as demonstrated by its Generation Chain-of-Thought, leading to a generated image that is more closely aligned with the prompt.

In parallel with advancements in visual generation, recent breakthroughs in language models have demonstrated that reinforcement learning (RL) can significantly enhance chain-of-thought reasoning capabilities. Models like OpenAI o1 (OpenAI, 2025) and DeepSeek-R1 (DeepSeek-AI, 2025) show that autoregressive language models can autonomously discover sophisticated reasoning strategies through self-improvement. Recognizing the natural synergy between autoregressive architectures and sequential reasoning, we introduce **GoT-R1**, a framework that adapts these RL advances to enhance semantic-spatial reasoning in autoregressive visual generation.

Extending reinforcement learning to enhance the reasoning abilities of autoregressive visual generation models presents unique challenges, unlike those encountered in code, mathematics, or traditional language tasks. First, designing appropriate rewards for visual generation is particularly challenging, as evaluating visual outputs requires assessing different dimensions: semantic fidelity to the prompt, accurate spatial arrangement of objects, proper binding of attributes to entities, coherence, and aesthetic quality. Second, optimizing solely on result rewards is suboptimal as it leaves the reasoning process unsupervised, potentially creating misalignments between the prompt, reasoning chain, and final image. Without explicit process supervision, the model may generate visually coherent but compositionally incorrect images, or fail to translate well-planned reasoning into accurate visual generation. Therefore, effective reinforcement learning for visual generation requires a comprehensive reward framework, evaluating both the reasoning process and the final output.

To address these challenges and inspired by the strong visual understanding and reasoning capabilities of multimodal large language models (MLLMs) (Bai et al., 2023; Liu et al., 2024b; OpenAI, 2025; Wang et al., 2025a), we leverage an MLLM-based base model for visual generation and propose a dual-stage Reinforcement Learning framework with unified MLLM-based multi-dimensional rewards. Our base generation model is an auto-regressive unified MLLM which takes text prompts as input and outputs the reasoning chain followed by a sequence of image tokens. Our reward model evaluates both the reasoning process and the final image output through a comprehensive set of reward signals: (1) *prompt-to-reasoning semantic alignment*, which assesses how well the reasoning chain captures the textual content; (2) *prompt-to-reasoning spatial alignment*, which evaluates the fidelity of planned spatial arrangements; (3) *reasoning-to-image alignment*, which measures how faithfully the generated image reflects the planned reasoning; and (4) *prompt-to-image alignment*, which evaluates the overall quality and compositional accuracy of the generated image.

We leverage MLLMs as reward models due to their ability to make nuanced judgments about text-image correspondence that align well with human assessments. We also enhance MLLMs' spatial evaluation capability by transforming bounding box coordinates into visualized bounding boxes drawn on a blank canvas, improving the reliability of the prompt-to-reasoning spatial reward. Through careful reward design and the adoption of Group Relative Policy Optimization (GRPO) (DeepSeek-AI, 2025), GoT-R1 enables models to autonomously discover effective reasoning strategies for complex visual scenes. Experimental results demonstrate significant improvements over the baseline model on T2I-CompBench and Geneval, advancing the state of compositional image generation. Figure 1 illustrates how GoT-R1 substantially improves the handling of compositional prompts. In summary, our main contributions are:

- We propose GoT-R1, a framework that enhances the semantic-spatial reasoning abilities for autoregressive visual generation by reinforcement learning, enabling models to discover effective reasoning strategies autonomously beyond predefined patterns.

- We design a comprehensive dual-stage multi-dimensional reward framework that evaluates both the intermediate reasoning process and final visual output from multiple perspectives, addressing the unique challenges of reinforcement learning for visual generation.

- We demonstrate significant performance improvements on the T2I-CompBench (Huang et al., 2023) and GenEval (Ghosh et al., 2023), particularly in compositional tasks requiring precise spatial relationships and attribute binding.

## 2 RELATED WORK

**Text-Driven Visual Generation**  Recent advancements in text-driven visual generation have been dominated by two main paradigms: diffusion models and autoregressive approaches. Diffusion models (Saharia et al., 2022; Rombach et al., 2022; Nichol et al., 2021; Ramesh et al., 2022; Zhang et al., 2023; Podell et al., 2023; Labs, 2024; Xie et al., 2024) have demonstrated remarkable success in generating high-fidelity images from text prompts by iteratively denoising an initial noise map. Autoregressive approaches (Sun et al., 2024a; Li et al., 2024; Tian et al., 2024; Han et al., 2024; Wang et al., 2024c; Yu et al., 2024; Wang et al., 2024b; Fang et al., 2023; Wang et al., 2025b), on the other hand, typically treat image generation as a sequence modeling problem. They often represent images as a sequence of discrete visual tokens (e.g., from a VQGAN) or patches and generate them element by element, commonly using large transformer architectures conditioned on textual input. Despite continuous improvements in generation quality, these methods still struggle with complex scenes involving complex text understanding, precise spatial relationships and attribute binding among multiple objects. Several studies have attempted to leverage large language models to enhance image generation capabilities. Models such as Chameleon (Team, 2024), Emu3 (Wang et al., 2024a), and Janus (Wu et al., 2024; Chen et al., 2025) explore unified architectures for visual understanding and generation. However, these approaches have yet to demonstrate that reasoning capabilities effectively translate to improved generation quality. Recently, GoT (Fang et al., 2025) introduced explicit semantic-spatial reasoning into image generations.

**Multimodal Large Language Models**  Multimodal Large Language Models (MLLMs)(Achiam et al., 2023; Bai et al., 2023; OpenAI, 2025) integrate vision encoders with LLMs, demonstrating strong visual understanding, sophisticated reasoning, and semantic analysis. Advanced MLLMs further enhance spatial understanding by grounding textual concepts to image regions(Liu et al., 2024b; Peng et al., 2023; Fang et al., 2024). However, despite unification attempts (e.g., Janus (Wu et al., 2024)) and models incorporating generation (e.g., Chameleon (Team, 2024), Emu2 (Sun et al., 2024b)), there remains a significant disconnect between understanding and generation capabilities. The rich semantic and spatial reasoning abilities of MLLMs are not yet fully leveraged in the generation process, as seen in models that generate images but may not fully utilize explicit semantic-spatial reasoning for synthesis.

**Reinforcement Learning for Reasoning**  Reinforcement Learning (RL) has emerged as a powerful approach for enhancing reasoning capabilities in large models. The success of OpenAI o1 (OpenAI, 2025) and DeepSeek-R1 (DeepSeek-AI, 2025)demonstrates how RL can significantly improve reasoning in language models. A notable algorithm contributing to some of these advancements is Group Relative Policy Optimization (GRPO) (Shao et al., 2024). GRPO is an efficient reinforcement learning technique that enhances policy learning by evaluating and normalizing rewards among a group of sampled candidate outputs from the model, eliminating the need for a separate critic model. Recent work has extended these techniques to multimodal domains. (Chen et al.; Deng et al., 2025; Liu et al., 2025; Yang et al., 2025; Zhang et al., 2025) Vision-R1 (Zhan et al., 2025) applies rule-based RL to enhance object localization in vision-language models without specialized reward models, using criterion-driven reward functions that evaluate completions based on visual feedback. Concurrent to our work, T2I-R1 (Jiang et al., 2025) introduces BiCoT-GRPO to jointly optimize semantic-level and token-level Chain-of-Thought reasoning for image generation, incorporating diverse vision experts as reward models.

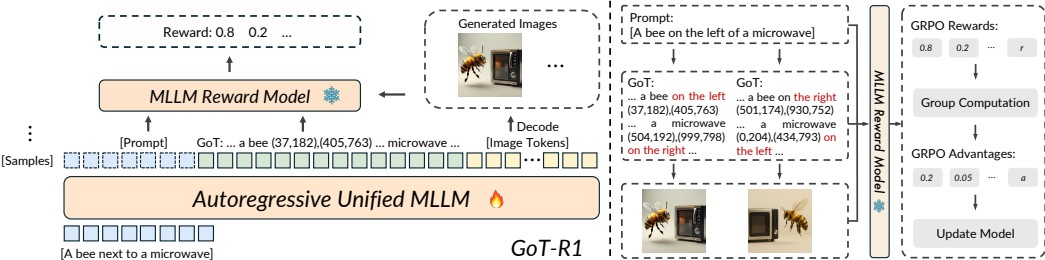

Figure 2: The GoT-R1 framework illustrating the reinforcement learning process with Group Relative Policy Optimization (GRPO). **Left**: Overview of the candidate sampling and initial evaluation stage, where diverse reasoning chains (GoT) and corresponding image tokens are generated from an input prompt, with an MLLM-based reward model providing preliminary scoring. **Right**: Detailed illustration of how MLLM-based rewards and advantages facilitate model updates via GRPO.

## 3 METHOD

In this section, we present the details of our GoT-R1 framework. We first review the prerequisite knowledge including the Generation Chain-of-Thought (GoT) paradigm and Group Relative Policy Optimization (GRPO) algorithm in Section 3.1. Then, we describe our GoT-R1 framework in Section 3.2, including the network architecture and training strategy. In Section 3.3, we elaborate on our MLLM-based dual-stage multi-dimensional reward design. The reward system thoroughly evaluates the alignment between prompt, reasoning, and generated image to provide comprehensive supervision signals for effective reinforcement learning.

### 3.1 PRELIMINARY

**Generation Chain-of-Thought (GoT)** Generation Chain-of-Thought (GoT) (Fang et al., 2025) is a paradigm that transforms visual generation through an explicit visual-semantic chain-of-thought reasoning process before outputting images. Unlike conventional text-to-image generation methods that directly map text embeddings to visual features, GoT decomposes complex prompts into a reasoning chain with both semantic descriptions and spatial coordinates. For example, given the prompt "A dog and a cat playing together", a GoT reasoning chain might include "a playful brown dog" with coordinates $(100, 200), (350, 450)$ and "an orange tabby cat" at $(400, 250), (650, 500)$, specifying both semantic attributes and spatial positioning of each object. This explicit chain-of-thought reasoning enables precise control over object attributes, spatial arrangements, and inter-object relationships, significantly improving compositional fidelity in the generated images.

In order to enable reasoning abilities of the generation model, GoT constructs large-scale training data with annotated reasoning chains following hand-crafted templates. The GoT framework is trained with the annotated data in a supervised manner to generate reasoning chains and images. However, this approach is inherently limited by the hand-crafted and fixed reasoning templates in training data, preventing the model from discovering more effective reasoning strategies. Moreover, the GoT framework trained with supervised fine-tuning tends to generate template-conformed but sometimes unfaithful reasoning chains, which can bottleneck subsequent visual generation.

**Group Relative Policy Optimization (GRPO)** Group Relative Policy Optimization (GRPO) is proposed by DeepSeek-R1 (Shao et al., 2024) to incentivize reasoning capabilities of large language models. It is an efficient RL algorithm that eliminates the need for a separate critic model. For each question $q$, GRPO samples a group of $G$ outputs $\{o_i\}_{i=1}^G$ from the current policy $\pi_{\theta_{\text{old}}}$. These outputs are evaluated using reward functions to obtain individual rewards $\{r_i\}_{i=1}^G$. The advantage for each sample is computed by normalizing the rewards within the group:

$$A_i = \frac{r_i - \text{mean}(\{r_j\}_{j=1}^G)}{\text{std}(\{r_j\}_{j=1}^G)} \tag{1}$$

The policy is then updated by optimizing the following objective:

$$J_{\text{GRPO}}(\theta) = \mathbb{E}_{q \sim \mathcal{D}, \{o_i\}_{i=1}^G \sim \pi_{\theta_{\text{old}}}(\cdot|q)}$$
$$\left[ \frac{1}{G} \sum_{i=1}^G \min\left(r_i(\theta)A_i, \text{clip}(r_i(\theta), 1-\epsilon, 1+\epsilon)A_i\right) - \beta D_{\text{KL}}(\pi_\theta || \pi_{\text{ref}}) \right] \tag{2}$$

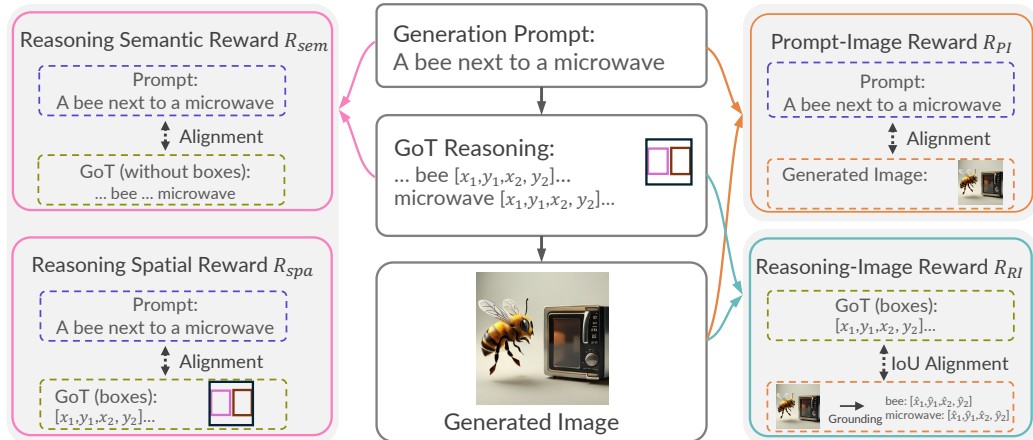

Figure 3: Overview of our MLLM-based dual-stage multi-dimensional reward framework. The diagram illustrates MLLM-based rewards assessing the intermediate GoT's semantic and spatial fidelity to the prompt, as well as the final image's alignment with both the prompt and the GoT.

where $r_i(\theta) = \frac{\pi_\theta(o_i|q)}{\pi_{\theta_{old}}(o_i|q)}$ is the probability ratio, $\epsilon$ is the clipping parameter, and $\beta$ controls the strength of the KL divergence penalty from a reference policy $\pi_{ref}$. This group-based approach provides a computationally efficient method for policy optimization while effectively leveraging relative performance differences within each group of samples.

## 3.2 GoT-R1 FRAMEWORK

GoT-R1 builds upon the Generation Chain-of-Thought (GoT) (Fang et al., 2025) framework for text-to-image generation by introducing reinforcement learning to enhance semantic-spatial reasoning capabilities. As discussed earlier, while GoT provides a strong foundation for compositional image generation, its effectiveness is limited by predefined reasoning templates in the training data. Our framework addresses this limitation by enabling the model to autonomously discover better reasoning strategies through reinforcement learning while maintaining the end-to-end optimization.

**Network Architecture** We adopt a unified MLLM that jointly models text and image tokens as our base architecture. For example, Janus-Pro (Chen et al., 2025) is capable of both visual understanding and generation tasks, processing images as discrete tokens alongside text with joint autoregressive modeling. This architecture allows us to generate textual reasoning chains and visual outputs in an end-to-end manner, enabling comprehensive optimization of the entire generation process.

**Training Strategy** Our base model has been trained on text-to-image generation task without chain-of-thought reasoning processes. To incentivize the reasoning abilities, our training process consists of two stages: In the first stage, we fine-tune the pre-trained model with reasoning chain and generated image annotations from GoT dataset. This stage of SFT establishes the basic capability to generate templated reasoning chains before generating image tokens, providing a strong initialization. In the second stage, we apply reinforcement learning to guide the model to explore free-style and more effective reasoning chains. For each prompt $P$, we sample $N$ different reasoning chains and corresponding images. These samples are then evaluated using our multi-dimensional reward function, which assesses both reasoning quality and generation fidelity. The model parameters are updated using GRPO to encourage high-reward reasoning strategies and generated images, and discourage the low-reward ones. The specific design of our reward function, which addresses the unique challenges of evaluating visual reasoning quality, is detailed in the following subsection.

## 3.3 MLLM-BASED DUAL-STAGE MULTI-DIMENSIONAL REWARD

The GoT-R1 generation framework is composed of two stages: prompt to reasoning chain generation, and reasoning chain to image generation. A straightforward integration with reinforcement learning would be to apply an end-to-end reward based solely on prompt-image alignment. However, without explicit constraints on the intermediate reasoning process, the reasoning chains may become unfaithful to the prompt or inconsistent with the final image, undermining the interpretability and

controllability of the generation pipeline. To guide the model toward faithful and consistent generation, we design a dual-stage reward mechanism with both result and intermediate process supervision. Specifically, we define four categories of rewards: (1) $R_{PI}$ measures the alignment between **Prompt** and generated **Image**, (2) $R_{PR}$ measures the faithfulness of **Reasoning** process to input **Prompt**, (3) $R_{RI}$ measures the fidelity of generated **Image** to **Reasoning** process and (4) $R_{HPS}$ which uses HPS v2.1 (Wu et al., 2023) to improve generation quality. For the prompt-to-reasoning alignment reward $R_{PR}$, we further decompose the reward into two distinct aspects—**semantic reward** $R_{sem}$ and **layout reward** $R_{spa}$—to ensure both the semantics and spatial arrangement in the reasoning process faithfully reflect the input prompt. All rewards are scaled to range [0,1]. We define total reward $R_{total}$ as the product of individual rewards:

$$R_{total} = R_{PI} * R_{PR} * R_{RI} * R_{HPS} = R_{PI} * \frac{(R_{sem} + R_{spa})}{2} * R_{RI} * R_{HPS} \qquad (3)$$

MLLMs are uniquely well-suited as reward models in this context due to their strong cross-modal understanding and reasoning capabilities. Trained on large-scale image-text pairs, MLLMs can provide unified, interpretable, and fine-grained evaluations for both reasoning chains and generated images across diverse aspects such as semantic consistency and spatial arrangement. This makes them ideal for reward functions in reinforcement learning settings, where conventional metrics often fall short in providing nuanced multi-dimensional feedback. The rewards are demonstrated in Fig. 3.

**Prompt-Image Reward** ($R_{PI}$)  The most intuitive reward design is the overall alignment between the input prompt and generated image. Leveraging the outstanding image understanding capabilities of MLLM, we utilize it to perform multi-dimensional evaluations of the final generated image, assessing whether it aligns with the composition (objects, attributes, layout etc.) specified in the prompt. The MLLM takes the input prompt and the generated image as input and predicts a discrete score ranging from 0 to 10 where 10 stands for the best.

**Prompt-Reasoning Semantic Reward** ($R_{sem}$)  To assess semantic consistency between the input prompt and generated GoT reasoning, we leverage MLLMs to evaluate each GoT in terms of missing elements (attributes), internal contradictions, logical consistency, and formatting quality. Specifically, the GoT reasoning along with the input prompt are input to MLLM to assess the reasoning chain from four dimensions with a score from 0 to 10: **1) Completeness**: Does the reasoning chain include all concepts mentioned in the prompt? **2) Faithfulness**: Does it introduce any content that contradicts the prompt? **3) Consistency**: Is the reasoning logically aligned with the described scene? **4) Clarity**: Is the content coherent and properly formatted?

**Prompt-Reasoning Spatial Reward** ($R_{spa}$) To evaluate the correctness of spatial planning by the reasoning chain, our MLLM reward model assesses whether the GoT object coordinates follow the spatial relationship (e.g., "left" or "top") from the prompt. However, lightweight LLMs or MLLMs exhibit limited sensitivity to bounding box coordinates and relationships between different spatial locations.

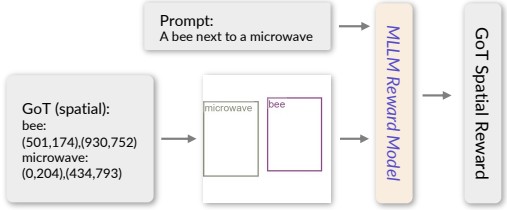

To bridge this capability gap, we propose an innovative MLLM-based layout evaluation approach based on a critical observation: MLLMs exhibit superior spatial comprehension when processing visual data compared to coordinates

Figure 4: Prompt-Reasoning Spatial Reward $R_{spa}$ process. For robust spatial evaluation, the MLLM assesses bounding boxes rendered on an image from the GoT's textual coordinates, rather than processing the coordinates directly as text.

in text form. Therefore, we convert textual coordinates into images by rendering corresponding bounding boxes on a blank canvas. With this visual format, the MLLM demonstrates significantly better spatial understanding and can provide clear and accurate scoring of the reasoning chain's spatial correctness. Figure 4 presents an illustration of this process.

**Reasoning-Image Reward** ($R_{RI}$)  During reinforcement learning, the model can occasionally generate images that deviate from its planned reasoning path. To further ensure that the GoT reasoning is faithfully reflected in the generated image, our framework incorporates an alignment reward between the GoT reasoning process and the generated image. Specifically, we expect each object planned in the GoT to appear at the corresponding location in the image. An MLLM is used to identify the location of each object in the generated image, yielding grounded bounding boxes denoted

as $B^{\text{Image}}$. For every object specified in GoT, we define its alignment reward as the Intersection over Union (IoU) between the planned bounding box ($B^{\text{GoT}}$) and its grounded counterpart in the image ($B^{\text{Image}}$). The overall reward $R_{RI}$ is then calculated as the average IoU across all N objects.

# 4 EXPERIMENT

## 4.1 TRAINING SETTINGS

We trained two models separately based on Janus-Pro-1B and Janus-Pro-7B (Chen et al., 2025). Our training process contains two stages: Pretraining on GoT-T2I dataset and online GRPO reinforcement learning with constructed prompt set. Specifically, We pretrain our model with LAHR-GoT (Schuhmann et al., 2022), JourneyDB-GoT (Sun et al., 2023) and FLUX-GoT datasets for 70000 steps, followed by 1000 steps of GRPO. Our constructed dataset for GRPO consists of prompts from T2I-Compbench training dataset and Laion-Aesthetics. When training with GRPO, the overall reward is calculated as the product of individual rewards described in Section 3.3. We employ low-rank adaptation (LoRA) (Hu et al., 2022) to efficiently update the trained model, with rank and lora alpha set to 32. Both phases operate end-to-end. In our GRPO training setup, we adopt a batch size of 8, a learning rate of $10^{-5}$, and employ a cosine learning rate schedule. For each input, we sample a group of $N = 16$ candidates and set both the text and image temperatures to 1.0. As the reward model, we adopt Qwen2.5VL-7B (Bai et al., 2025). The loss is computed over the entire generated output sequence. GRPO training was conducted on 8 NVIDIA L40S GPUs in 48 hours.

## 4.2 QUANTITATIVE EVALUATION

In this section, we present the quantitative results of our model on numerous benchmarks.

**T2I-CompBench Results**.Table 1 shows our model's performance compared to three categories of approaches: diffusion models with frozen encoders, two-stage layout-guided models, and autoregressive models enhanced with LLMs/MLLMs. GoT-R1-7B achieves state-of-the-art results, obtaining the highest scores in five of six evaluation categories with up to 15% improvement after 1000 GRPO fine-tuning steps. Our model shows notable performance on the Complex compositions category, while GoT-R1-1B outperforms larger models like Janus-Pro-7B in several categories.

**GenEval Results**.On the GenEval benchmark (Table 2), GoT-R1-7B establishes a new state-of-the-art with an overall score of 0.75, representing improvements over the base Janus-Pro-GoT-7B model. The performance gains are particularly notable in compositional abilities: two-object generation improves from 0.69 to 0.94, and attribute binding advances from 0.43 to 0.68. These results demonstrate the effectiveness of combining structured reasoning with reinforcement-guided optimization for complex visual generation tasks.

**General Image Quality Results.** To assess performance beyond compositional benchmarks, we evaluate on the COCO 2014 validation set (30k images) using standard image quality metrics. Table 4 shows our model achieves improvements in CLIP Score and Aesthetic Score. Human evaluation on 300 randomly selected prompts demonstrates strong preference for GoT-R1-7B (77%) over baseline models, indicating enhanced overall generation quality alongside compositional accuracy.

## 4.3 QUALITATIVE EVALUATION

Figure 5 presents a qualitative comparison among the base model Janus-Pro-7B, the GoT-finetuned model Janus-Pro-7B-GoT, and our GRPO-enhanced model GoT-R1-7B. We showcase examples generated from compositional prompts involving multiple attributes, relative spatial relationships, and object numeracy.

Table 4: General image quality evaluation on COCO 2014 validation set.

| Model | CLIP↑ | Aesthetic↑ | Human↑ |
|---|---|---|---|
| Janus-Pro-7B | 28.67 | 4.88 | 9% |
| Janus-Pro-GoT-7B | 29.97 | 5.19 | 14% |
| **GoT-R1-7B** | **31.83** | **5.41** | **77%** |

While the GoT-finetuned model produces images of higher quality than the base model, it still struggles with complex compositional generation. In contrast, GoT-R1-7B demonstrates stronger prompt alignment, accurately reflecting even unnatural prompts in its generations. In addition, GoT-R1-7B generates detailed and aesthetically appealing visual contents. These gains are largely attributed to our MLLM-based reward design, which guides the model to optimize both semantic and spatial alignment across the GoT reasoning process and output image. By leveraging fine-grained evalua-

Table 1: Quantitative evaluation of text-to-image generation on T2I-CompBench. GoT models refer to Janus-Pro finetuned using the GoT framework, while GoT-R1 models denote further training via GRPO on the GoT-finetuned checkpoints. GoT-R1 models are evaluated under guidance scale 5.

| Model | Color | Shape | Texture | 2D-Spatial | Non-Spatial | Complex |
|---|---|---|---|---|---|---|
| *Diffusion Models* | | | | | | |
| SD-v1.5 (Rombach et al., 2022) | 0.3758 | 0.3713 | 0.4186 | 0.1165 | 0.3112 | 0.3047 |
| SD-XL-base-1.0 (Podell et al., 2023) | 0.5879 | 0.4687 | 0.5299 | 0.2131 | 0.3119 | 0.3237 |
| DALLE·3 (Betker et al., 2023) | 0.7785 | **0.6205** | 0.7036 | 0.2865 | 0.3003 | 0.3773 |
| Stable v3 (Esser et al., 2024) | 0.8132 | 0.5885 | 0.7334 | 0.3200 | 0.3140 | 0.3771 |
| FLUX.1 (Labs, 2024) | 0.7407 | 0.5718 | 0.6922 | 0.2863 | 0.3127 | 0.3703 |
| *Layout Guided Two-stage Models* | | | | | | |
| Ranni (Feng et al., 2024) | 0.6893 | 0.4934 | 0.6325 | 0.3167 | - | - |
| LayoutGPT-Llama7B (Feng et al., 2023) | 0.3296 | 0.3654 | 0.3982 | 0.1443 | 0.2990 | 0.2768 |
| *Auto-regressive Models* | | | | | | |
| Emu3 (Wang et al., 2024a) | 0.7544 | 0.5706 | 0.7164 | - | - | - |
| Janus-Pro-1B (Chen et al., 2025) | 0.3411 | 0.2261 | 0.2696 | 0.0968 | 0.2808 | 0.2721 |
| Janus-Pro-1B-GoT | 0.6336 | 0.4456 | 0.5621 | 0.2140 | 0.3070 | 0.3490 |
| GoT-R1-1B | 0.7632 | 0.5174 | 0.6589 | 0.2674 | 0.3101 | 0.3749 |
| Janus-Pro-7B (Chen et al., 2025) | 0.6359 | 0.3528 | 0.4936 | 0.2061 | 0.3085 | 0.3559 |
| Janus-Pro-7B-GoT | 0.6551 | 0.5008 | 0.5836 | 0.2457 | 0.3113 | 0.3754 |
| **GoT-R1-7B** | **0.8139** | 0.5549 | **0.7339** | **0.3306** | **0.3169** | **0.3944** |

Table 2: Quantitative evaluation on GenEval (Ghosh et al., 2023). Obj.: Object. Attr.: Attribution.

| Method | Architecture | Overall | Single Obj. | Two Obj. | Counting | Colors | Position | Attr. Binding |
|---|---|---|---|---|---|---|---|---|
| SDv1.5 (Rombach et al., 2022) | Diffusion | 0.43 | 0.97 | 0.38 | 0.35 | 0.76 | 0.04 | 0.06 |
| SDv2.1 (Rombach et al., 2022) | Diffusion | 0.50 | 0.98 | 0.51 | 0.44 | 0.85 | 0.07 | 0.17 |
| SD-XL (Podell et al., 2023) | Diffusion | 0.55 | 0.98 | 0.74 | 0.39 | 0.85 | 0.15 | 0.23 |
| DALLE-2 (Ramesh et al., 2022) | Diffusion | 0.52 | 0.94 | 0.66 | 0.49 | 0.77 | 0.10 | 0.19 |
| SD3 (d=24) (Esser et al., 2024) | Diffusion | 0.62 | 0.98 | 0.74 | 0.63 | 0.67 | 0.34 | 0.36 |
| LayoutGPT (Feng et al., 2023) | Two-stage | 0.41 | 0.97 | 0.51 | 0.26 | 0.56 | 0.11 | 0.07 |
| RPG (Yang et al., 2024) | Two-stage | 0.50 | 0.98 | 0.66 | 0.17 | 0.85 | 0.07 | 0.27 |
| LlamaGen (Sun et al., 2024a) | Autoregressive | 0.32 | 0.71 | 0.34 | 0.21 | 0.58 | 0.07 | 0.04 |
| Chameleon (Team, 2024) | Autoregressive | 0.39 | - | - | - | - | - | - |
| LWM (Liu et al., 2024a) | Autoregressive | 0.47 | 0.93 | 0.41 | 0.46 | 0.79 | 0.09 | 0.15 |
| SEED-X (Ge et al., 2024) | MLLM+Diffusion | 0.49 | 0.97 | 0.58 | 0.26 | 0.80 | 0.19 | 0.14 |
| Emu3-Gen (Wang et al., 2024a) | Autoregressive | 0.54 | 0.98 | 0.71 | 0.34 | 0.81 | 0.17 | 0.21 |
| Janus (Wu et al., 2024) | Autoregressive | 0.61 | 0.97 | 0.68 | 0.30 | 0.84 | 0.46 | 0.42 |
| JanusFlow (Ma et al., 2024) | Autoregressive | 0.63 | 0.97 | 0.59 | 0.45 | 0.83 | **0.53** | 0.42 |
| GoT (Fang et al., 2025) | MLLM+Diffusion | 0.64 | **0.99** | 0.69 | **0.67** | 0.85 | 0.34 | 0.27 |
| Janus-Pro-7B-GoT | Autoregressive | 0.64 | **0.99** | 0.69 | 0.48 | 0.85 | 0.43 | 0.43 |
| **GoT-R1-7B** | Autoregressive | **0.75** | **0.99** | **0.94** | 0.50 | **0.90** | 0.46 | **0.68** |

Table 3: Ablation study on reward design. All models are trained for 1000 steps using GRPO based on the Janus-Pro-1B-GoT (Baseline). Evaluations are conducted with a guidance scale of 5.

| Method | $R_{sem}$ | $R_{spa}$ | $R_{RI}$ | $R_{PI}$ | Color | Shape | Texture | 2D-Spatial | Non-Spatial | Complex |
|---|---|---|---|---|---|---|---|---|---|---|
| Baseline | × | × | × | × | 0.6336 | 0.4456 | 0.5621 | 0.2140 | 0.3070 | 0.3490 |
| w$R_{PR}$ | ✓ | ✓ | × | × | 0.7050 | 0.4671 | 0.6075 | 0.2283 | 0.3089 | 0.3619 |
| w$R_{RI}$ | × | × | ✓ | × | 0.3340 | 0.2563 | 0.3940 | 0.0076 | 0.2537 | 0.2488 |
| w$R_{PI}$ | × | × | × | ✓ | 0.7401 | 0.5066 | 0.6308 | 0.2398 | 0.3076 | 0.3724 |
| w$R_{PR}$&$R_{PI}$ | ✓ | ✓ | × | ✓ | 0.7289 | 0.4893 | 0.6485 | 0.2557 | 0.3094 | 0.3653 |
| w$R_{PR}$&$R_{RI}$ | ✓ | ✓ | ✓ | × | 0.7118 | 0.4582 | 0.6243 | 0.2579 | 0.3097 | 0.3583 |
| w$R_{RI}$&$R_{PI}$ | × | × | ✓ | ✓ | 0.6507 | 0.4299 | 0.5913 | 0.1797 | 0.3010 | 0.3452 |
| w$R_{sem}$ | ✓ | × | ✓ | ✓ | 0.7323 | 0.4729 | 0.6251 | 0.2133 | 0.3094 | 0.3568 |
| w$R_{spa}$ | × | ✓ | ✓ | ✓ | 0.7067 | 0.4685 | 0.6115 | 0.2419 | 0.3089 | 0.3648 |
| **GoT-R1-1B** | ✓ | ✓ | ✓ | ✓ | **0.7632** | **0.5174** | **0.6589** | **0.2674** | **0.3101** | **0.3749** |

tions from MLLM, our reward formulation enables GoT-R1-7B to excel not only in visual quality but also in faithfully capturing the intent of complex prompts.

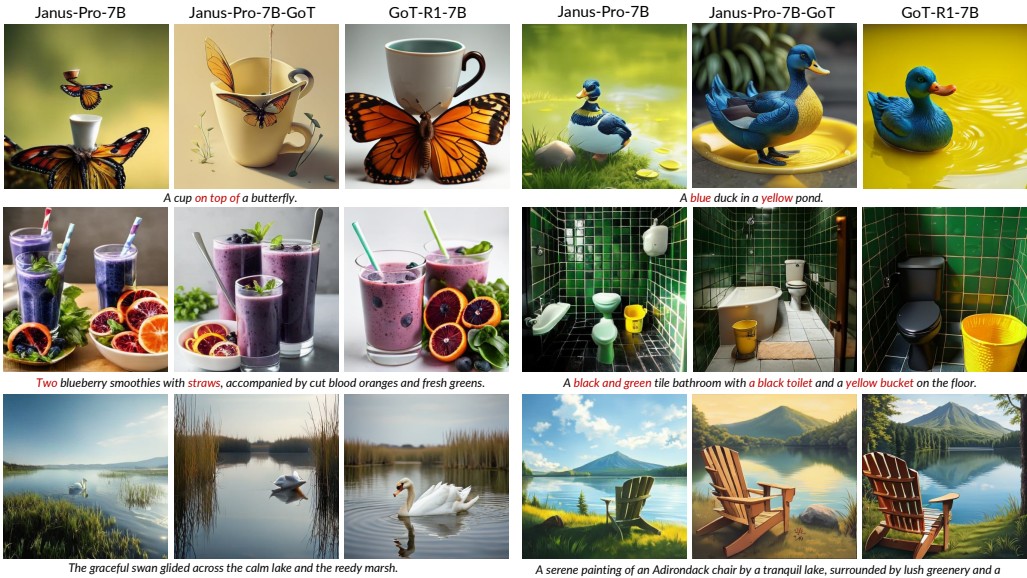

Figure 5: Qualitative comparison among the base model Janus-Pro-7B, the GoT-finetuned checkpoint Janus-Pro-7B-GoT, and our GRPO-enhanced model GoT-R1-7B. Our model demonstrates superior performance on prompt alignment and image quality.

## 4.4 ANALYSIS ON SELF-EXPLORED GENERATION CHAIN-OF-THOUGHT

To assess the quality of reasoning, we compared the self-explored Generation Chain-of-Thought from GoT-R1-7B against the predefined GoT of Janus-Pro-7B-GoT. GPT-4o (Achiam et al., 2023) evaluated the GoT content for 100 prompts randomly sampled from each of T2I-CompBench's Color, Spatial, and Complex categories, plus 100 from LAION-5B (Schuhmann et al., 2022). Voting was based on four criteria: **relevance to the input prompt**, **accuracy of object descriptions and bounding boxes**, and **the clarity and fluency of the text**. As detailed in Table 5, GoT-R1-7B's self-explored reasoning is overwhelmingly preferred by GPT-4o across all evaluated categories. This strong preference underscores GoT-R1's ability to autonomously discover and generate superior reasoning paths, a key factor contributing to its enhanced compositional generation capabilities.

## 4.5 ABLATION STUDY ON REWARD DESIGN

We conduct a thorough ablation study on our MLLM-based dual-stage multi-dimensional reward and key training settings to validate their contributions. All experiments are performed on T2I-CompBench, and trained for 1000 steps using GRPO

Table 5: GPT-4o vote results comparing Janus-Pro-7B-GoT and GoT-R1-7B on reasoning quality.

| Method | Color | Spatial | Complex | LAION-5B |
|---|---|---|---|---|
| Janus-Pro-7B-GoT | 21 | 16 | 29 | 31 |
| **GoT-R1-7B** | **79** | **84** | **71** | **69** |

based on the Janus-Pro-1B-GoT checkpoint, which serves as our baseline. Results are displayed in Table 3 and 6.

**Ablation Study on Reward Design** Table 3 reveals both the contribution and limitation of each reward component. Training with only $R_{PI}$ yields the best performance among these single-reward variants but still falls short of the full GoT-R1-1B, as the

Table 6: Ablation study on training details. We present results on T2I-Compbench evaluated under guidance scale 5.

| Method | Color | Shape | Texture | Spatial | Non-Spatial | Complex |
|---|---|---|---|---|---|---|
| Baseline | 0.6336 | 0.4456 | 0.5621 | 0.2140 | 0.3070 | 0.3490 |
| Sum reward | 0.7154 | 0.4385 | 0.5608 | 0.2254 | 0.3080 | 0.3638 |
| Text evaluated $R_{spa}$ | 0.7166 | 0.4289 | 0.6311 | 0.2158 | 0.3098 | 0.3554 |
| Conventional rewards | 0.5914 | 0.4284 | 0.5607 | 0.1388 | 0.2936 | 0.3353 |
| GoT-R1 | **0.7632** | **0.5174** | **0.6589** | **0.2674** | **0.3101** | **0.3749** |

GoT reasoning process is largely bypassed. Relying solely on $R_{PR}$ leads to poorer outcomes, underscoring the necessity of rewarding the final generated image. Furthermore, using only $R_{RI}$ can be detrimental, because the absence of prompt-reasoning reward $R_{PR}$ results in a misaligned reasoning process and thus provides harmful guidance to image generation. Further experiments

in Table 3, where individual reward components are removed from our full reward set, reinforce this conclusion. Removing either $R_{RI}$ or $R_{PI}$ leads to a noticeable degradation in performance. Critically, removing $R_{PR}$ while retaining $R_{RI}$ once again results in more significant performance decline, as the model attempts to align the image with potentially flawed reasoning. These findings collectively justify the importance of our comprehensive reward design that aligns all stages of the generation process.

**Ablation Study on $R_{PR}$ Composition** Regarding the composition of $R_{PR}$, we ablate its two constituents, $R_{sem}$(prompt-reasoning semantic reward) and $R_{spa}$ (prompt-reasoning spatial reward), by training models where only one is active. The results in Table 3 demonstrate their complementary roles: $R_{sem}$ primarily enhances attribute binding, whereas $R_{spa}$ improves spatial consistency, confirming the value of their combination within $R_{PR}$.

**Ablation Study on Training Details** We further ablate three key settings in our training. In Equation 3, the total reward $R_{total}$ is the product of its individual terms. We evaluate an alternative setting that sums the rewards instead. Moreover, we ablate our novel MLLM layout evaluation approach, where instead of converting GoT layout plans to image for MLLM to assess, $R_{spa}$ is given by MLLM evaluating GoT layout directly from its textual coordinates. Last but not least, we replace all MLLM-based rewards with conventional metrics: CLIP similarity for the prompt-image reward and Grounding DINO (Liu et al., 2024c) for the reasoning-image alignment. The results are presented in Table 6. The findings affirm the efficacy of our training settings in optimizing GoT-R1.

## 5 CONCLUSION

In this paper, we introduce GoT-R1, a novel framework that enhances the semantic-spatial reasoning capabilities of autoregressive visual generation models through reinforcement learning. Building on the Generation Chain-of-Thought paradigm, GoT-R1 empowers models to move beyond predefined templates and autonomously discover more effective reasoning strategies for complex, compositional prompts. A key contribution of our work is the development of a MLLM-based dual-stage multi-dimensional reward framework which evaluates both the intermediate reasoning process and the final visual output, assessing semantic alignment, spatial accuracy, and overall image quality. Our experimental results on the T2I-CompBench and GenEval benchmarks demonstrate significant improvements over existing methods, particularly in tasks requiring precise attribute binding and spatial relationships. GoT-R1 advances the state-of-the-art and opens new avenues for creating more accurate and contextually aware visual content. We leverage GPT-5 for grammar refinement.

## 6 ETHICS AND REPRODUCIBILITY STATEMENT

Our work builds on publicly available datasets such as T2I-CompBench trainset and GoT datasets, without the use of private or sensitive data. We are mindful that text-to-image generation can be misused for creating misleading or harmful content; thus, our experiments are limited to research benchmarks and synthetic evaluation only. To ensure reproducibility, we report all model architectures, training setups, and hyperparameters in detail (Sections 3–4), and validate robustness through multiple runs (in appendix) and ablation studies. Upon acceptance, we will release code, checkpoints, and evaluation scripts to support transparent verification and further research.

## 7 ACKNOWLEDGMENT

This work is supported by the National Nature Science Foundation of China (No. 62402406).

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

# A  QUALITATIVE EVALUATION

We present more qualitative analysis on our GoT-R1-7B model in Figure 6. This figure show-cases a comparison of text-to-image generation capabilities among the baseline Janus-Pro-7B, the GoT-finetuned Janus-Pro-GoT-7B, and our GoT-R1-7B model across various prompts, highlighting differences in image quality and prompt adherence.

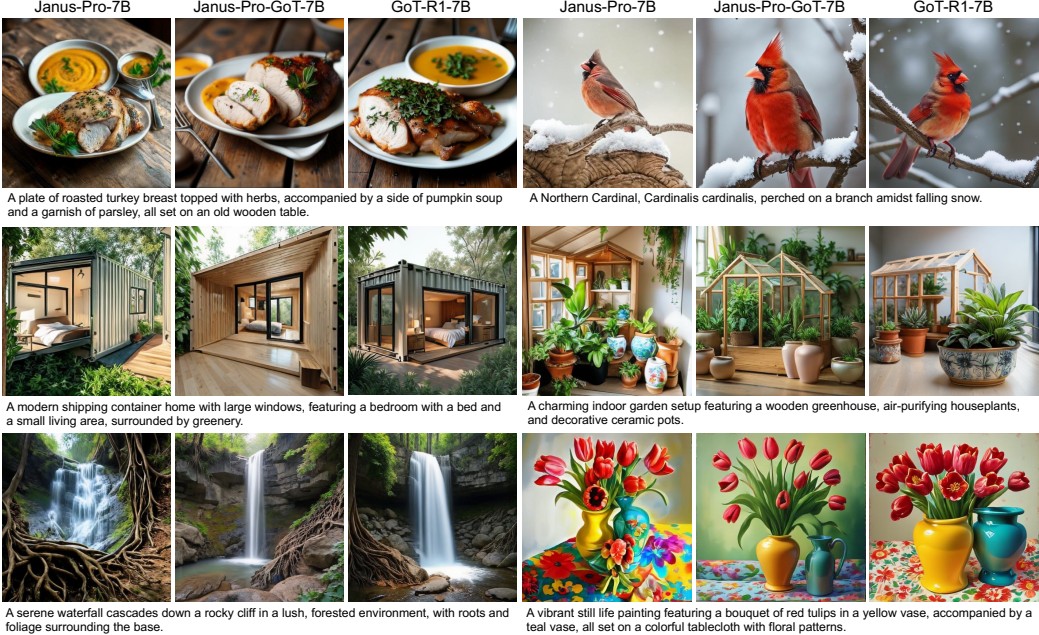

Figure 6: Samples of text-to-image generation by Janus-Pro-7B, Janus-Pro-GoT-7B and GoT-R1-7B.

# B  ADDITIONAL QUANTITATIVE ANALYSIS

**MLLM Reward Model Reliability.** To validate our reward model choice, we measured correlation with human judgment using Kendall's tau and Spearman's rho on 500 T2I-CompBench prompt-image pairs. Table 7 shows our Qwen2.5VL-7B reward model achieves higher alignment with human preference compared to CLIP Score.

Table 7: Correlation analysis with human judgment for different reward models.

| Model | Kendall's tau ($\tau$) | Spearman's rho ($\rho$) |
|---|---|---|
| CLIP Score | 0.1810 | 0.2141 |
| **Qwen2.5VL-7B (Ours)** | **0.3147** | **0.3428** |

**Training Stability.** Three independent training runs from different random seeds show consistent performance across all T2I-CompBench categories, with variance less than ±0.02 as shown in Table 8, confirming the robustness of our training process.

**Multi-dimensional Reward Design.** An experiment using only HPSv2 as reward signal demonstrates the importance of our multi-component design. Table 9 shows that while achieving higher HPSv2 scores, the single-reward model exhibits reward hacking with degraded compositional performance, confirming that our balanced reward formulation prevents optimization pathologies.

Table 8: Training stability analysis across multiple runs on T2I-CompBench.

| Category | Run 1 | Run 2 | Run 3 |
|---|---|---|---|
| Color | 0.8139 | 0.8097 | 0.8211 |
| Shape | 0.5549 | 0.5610 | 0.5633 |
| Texture | 0.7339 | 0.7341 | 0.7450 |
| 2D-Spatial | 0.3306 | 0.3398 | 0.3289 |
| Complex | 0.3944 | 0.3895 | 0.3937 |

Table 9: Comparison of single vs. multi-dimensional reward training.

| Method | HPSv2 Score ↑ | CLIP Score ↑ | 2D-Spatial | Complex |
|---|---|---|---|---|
| HPSv2 Only | **28.9** | 25.3 | 0.1891 | 0.2847 |
| GoT-R1 (Multi-reward) | 27.2 | **31.83** | **0.3306** | **0.3944** |

## C  LIMITATIONS

Our approach has several constraints that should be acknowledged.

**Position Prediction in Object-Heavy Scenes:** The model's ability to generate accurate and corresponding bounding boxes for each distinct object decreases when prompts contain **more than ten items**. This can impact the fidelity of very complex scenes.

**Inference Latency:** We acknowledge that the explicit generation of the GoT reasoning chain introduces additional latency during inference. Our internal benchmarks show this can increase the total generation time by **approximately 30%** compared to direct generation without the reasoning step.

**Reliance on the Reward Model's Capability:** The performance of our framework is inherently tied to the power of the MLLM used as a reward function. While our results show the current reward model is effective, we acknowledge that **training GoT-R1 with an even more powerful, next-generation MLLM could further unlock performance gains**. This reliance represents a dependency on the progress of external models and is a key avenue for future improvement.

## D  MLLM-BASED REWARD EVALUATION PROMPTS

We present the prompt used in our paper in Figure[ 8, 9, 10, 11, 12]. Specifically, Figure 8 details the prompt used for evaluating the semantic consistency between prompt and reasoning chain. Figure 9 shows the prompt for evaluating the spatial layout predicted in reasoning chain. Figure 10 displays the assessment prompt for prompt-image alignment. Figure 11 illustrates the prompt used for grounding in the reasoning-image reward. Figure 12 provides the prompt utilized for comparing reasoning chains with GPT-4o.

## E  REWARD CURVE IN GRPO TRAINING

Figure 7 shows the total reward curve during GRPO training. We rescale the reward values such that the reward at step 0 is mapped to 0, and the reward at step 1000 is mapped to 1 for better visualization. The reward steadily increases over the 1000 training steps and gradually stabilizes, indicating that the model consistently learns higher-quality reasoning and generation strategies under the multi-dimensional reward framework.

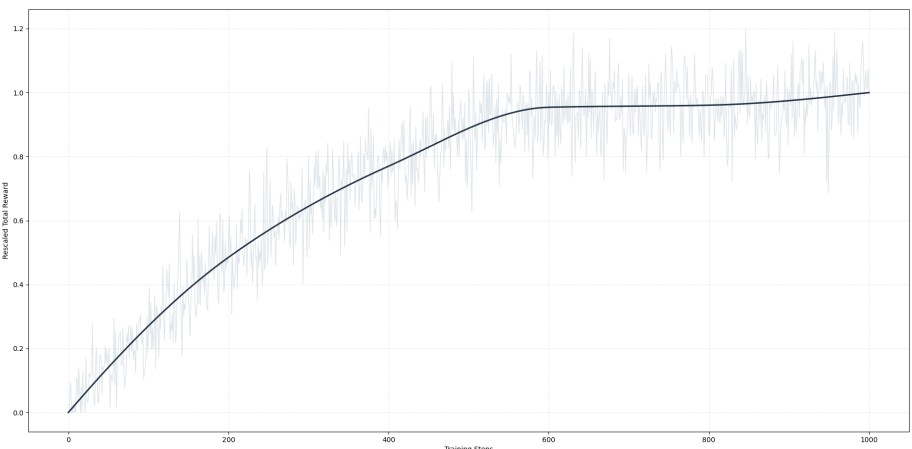

Figure 7: Total reward curve of our GRPO stage. We rescale the reward values such that the reward at step 0 is mapped to 0, and the reward at step 1000 is mapped to 1 for better visualization.

**Human:** You are a professional image caption evaluator. You will evaluate how well a detailed AI-generated caption aligns with a brief image prompt.
You will be given:
1. A brief image prompt that describes what should be in the image
2. A detailed caption that was generated based on the brief prompt
Your task is to evaluate if the detailed caption is aligned with and faithful to the brief prompt.
Consider:
- Does the detailed caption include all elements from the brief prompt?
- Does the detailed caption add elements that contradict the brief prompt?
- Is the detailed caption reasonable and consistent with what the prompt describes?
- Is the caption coherent and properly formatted?
The score should be from 0 to 10:
- 0: Completely nonsensical output, messy code, or gibberish that fails to function as a caption
- 1-2: Severe misalignment. The detailed caption fails to represent key elements or completely contradicts the brief prompt
- 3-4: Poor alignment with significant omissions or contradictions to the brief prompt
- 5-6: Moderate alignment with some missing elements or noticeable inconsistencies
- 7-8: Strong alignment with minor inconsistencies or additions that don't contradict the prompt
- 9-10: Perfect alignment. The detailed caption faithfully includes all elements from the brief prompt with appropriate elaboration
Brief prompt: **Prompt**
Detailed caption: **Reasoning Chain**
Note to only ouput with a dictionary with score in this format: {"score": ...}
**Assistant:**

Figure 8: Prompt for $R_{sem}$ evaluation.

**Human:** Determine if objects are arranged as described in the prompt by analyzing the image.
ORIGINAL IMAGE PROMPT: **prompt**
COORDINATE SYSTEM EXPLANATION:
- The image shows object bounding boxes with names labeled at the top-left corner of each box
SCORING RULES:
- Score 8-10 if the objects are shown in the image and their positions MATCH the relationship in the prompt
* 10: Perfect match with clear relationship
* 9: Strong match with minor ambiguity
* 8: Good match with some ambiguity
- Score 5-7 if the relationship is partially correct or ambiguous
* 7: Mostly correct with some misalignment
* 6: Relationship is ambiguous but leaning toward correct
* 5: Borderline case where relationship could be interpreted either way
- Score 1-4 if the objects are NOT shown in the image or positions CONTRADICT the relationship in the prompt
* 4: Slight contradiction or missing one object
* 3: Clear contradiction but objects are present
* 2: Major contradiction or missing multiple objects
* 1: Complete mismatch with the prompt
Please answer in order to: Verify if the objects are shown in the ORIGINAL IMAGE PROMPT.
Decide if the relationships between objects match what is described in the ORIGINAL IMAGE PROMPT.
Your response MUST be formatted as:
{{
"reasoning": ...,
"score": ...
}}
Output only the dictionary with nothing else.
$< Image >$ **Visualized reasoning chain** $< /Image >$
**Assistant:**

Figure 9: Prompt for $R_{spa}$ evaluation.

**Human:**
You are an expert in visual analysis specializing in compositional accuracy evaluation.
Your task is to compare the caption with an image and assess ONLY how well the image matches the described elements, objects, and their relationships.
Analyze:
Compositional accuracy: Evaluate if all key elements mentioned in the caption appear in the image with correct relationships, positioning, and attributes as specified.
EVALUATION CRITERIA:
1. Object Presence: Are the key objects mentioned in the image prompt correctly shown in the image?
2. Spatial Positioning: Are the objects positioned in the EXACT spatial relationships described in the caption? Pay special attention to positional terms like "on top of," "next to," "inside," "left of," "right of," "behind," "in front of," etc.
Examples of STRICT spatial interpretations:
- "Left of" means the object must be positioned horizontally to the left, not above, below, or on top.
- "On top of" means the object must be directly above and touching, not beside or below.
Compositional accuracy score (0-10):
- 8-10: Perfect match. Image contains all elements with EXACTLY the spatial relationships described.
- 5-7: Minor mismatch. All objects present but with slightly incorrect spatial relationships.
- 0-4: Major mismatch. Objects present but with completely incorrect spatial relationships, or missing key objects.
Caption: **prompt**
Your response MUST be formatted as:
{{
"description": "ONE sentence describing the image accurately, including the spatial relationship observed",
"caption": "repeat of the image caption provided",
"reasoning": "ONE sentence explaining if the spatial positioning in the image EXACTLY matches or contradicts the caption",
"score": ...
}}
Output only the python dictionary with nothing else.
$< Image >$ **Generated Image** $< /Image >$
**Assistant:**

Figure 10: Prompt for $R_{PI}$ evaluation.

**Human:**
Locate the $< object >$, report the bbox coordinates in JSON format.
**Assistant:**

Figure 11: Prompt for $R_{RI}$ grounding.

**Human:**
You are an assistant tasked with evaluating two detailed image captions based on a given input prompt. Your goal is to determine which caption provides a better and more accurate description of the image, considering the object descriptions and their corresponding positions.
Task: Evaluate the two detailed image captions provided below, based on the given input prompt. Select the caption that is a better and more accurate description of an image, considering the object descriptions and their corresponding bounding boxes. The detailed captions includes the bounding boxes of the objects in the image, which are represented as (x1, x2), (y1, y2). (Assume a standard image coordinate system where (0,0) is the top-left corner).
Input Prompt:
**prompt**
Detailed Caption A:
**Reasoning Chain A**
Detailed Caption B:
**Reasoning Chain B**
When deciding which caption is better, please consider the following:
Relevance to the Input Prompt: How well does each caption address and align with the original input prompt?
Accuracy of Object Descriptions: Are the objects described correctly and in sufficient detail?
Accuracy of Bounding Boxes: Do the provided bounding boxes (x1, x2), (y1, y2) accurately represent the location and extent of the described objects?
Completeness: Does the caption identify and describe the key objects relevant to the input prompt? Does it miss any important elements or include irrelevant ones?
Clarity and Coherence: Is the caption easy to understand? Are the object descriptions and their spatial relationships (implied by bounding boxes) presented logically?
Naturalness and Fluency: Does the caption read like a natural and well-written description?
Specificity vs. Generality: Does the caption provide an appropriate level of detail based on the input prompt, or is it too vague or overly specific?
Output Format:
Please provide your response in the following format:
{{
Reasoning: "Your reasoning here",
Selected Caption: "A or B",
}}
**Assistant:**

Figure 12: Prompt for GPT-4o reasoning chain comparison.

