# OpenReview forum: "GoT-R1: Unleashing Reasoning Capability of Autoregressive Visual Generation with Reinforcement Learning"
_ICLR.cc/2026/Conference — ICLR 2026 Poster_

### Official Review · Reviewer_bbst · 2025-10-22

**Soundness:** 3
**Presentation:** 4
**Contribution:** 2
**Rating:** 2
**Confidence:** 3

**Summary:**

This paper proposes GoT-R1, an RL framework to improve autoregressive image generation models on complex text prompts by incorporating explicit reasoning. The approach introduces a dual-stage, multi-dimensional reward scheme: one stage evaluates the reasoning process itself, and another evaluates the final image, using vlm as judges. On multiple benchmarks, GoT-R1 shows significant improvements in placing objects correctly and binding attributes, outperforming the previous sota in complex prompt fidelity.

**Strengths:**

* The method shows significant improvement on compositionally of image gen tasks.
* Clear ablation study that supports the dual stage and reward design, RPI‑only helps but full reward performs best, while RRI‑only is harmful

**Weaknesses:**

* The idea of injecting chain-of-thought into image generation and refining with RL is not unique to this paper. For example: ReasonGen-R1, T2I-R1 are both very similar works and comparative discussions are limited.
* The experiments seem mostly focused on the compositional benchmark, evaluation on broader text-to-image tasks are very limited.

**Questions:**

For equation (3), the total reward is a sum (R_sem + R_spa) which is [0,2] times a bunch of other [0,1]s, wouldn't this make the total reward be [0,2]?

---

> ### Author Response · Authors · 2025-11-19
> **Rebuttal response to Reviewer bbst**
>
> We sincerely thank the reviewer for their valuable questions. We appreciate this opportunity to clarify the novelty of our work and the breadth of our evaluations.
>
> **Weakness 1: The idea of injecting chain-of-thought into image generation and refining with RL is not unique to this paper. For example: ReasonGen-R1, T2I-R1 are both very similar works and comparative discussions are limited.**
>
> We thank the reviewer for raising this important point. We would first like to gently clarify that **T2I-R1 and ReasonGen-R1 are concurrent works, as none were accepted at the time of our submission**.
>
> Moreover, our **GoT-R1** framework is fundamentally different in its design philosophy and technical implementation, which we believe is our key contribution.
>
> 1. Fundamentally distinguishing GoT-R1 from both ReasonGen-R1 and T2I-R1 is our **comprehensive reward supervision** and the **semantic-spatial nature** of our reasoning. Both ReasonGen-R1 and T2I-R1 rely primarily on **result-based supervision**, calculating rewards solely based on the final generated image. However, our ablation study (Table 4) empirically demonstrates that this approach is insufficient because it leaves the intermediate reasoning step unsupervised and may provides reasoning chains unfaithful to the prompt. GoT-R1 addresses this by supervising the entire generation pipeline (Prompt, Reasoning and Image) which is essential for preventing intermediate errors and ensuring robust performance. Moreover, GoT-R1 includes spatial bounding boxes in the reasoning chain and employs a dedicated spatial reward ($R_{spa}$) to evaluate layout accuracy. This contrasts with the purely textual rationales used in concurrent works, which lack this explicit mechanism for spatial planning and supervision.
>
> 2. In technical implementation, our approach offers a more unified and robust reward design compared to T2I-R1. T2I-R1 relies on a complex "ensemble of diverse vision experts" to calculate rewards. In contrast, we propose a unified reward framework, demonstrating that a single, powerful MLLM (Qwen2.5-VL) can reliably serve as a comprehensive judge for semantic, spatial, and image alignment .
>
>    More importantly, **T2I-R1 employs VQA models as reward signals , which mirror T2I-CompBench's own evaluation metrics.** This creates a high risk, as the model is effectively optimized directly on the test metric. In contrast, GoT-R1 decouples training from evaluation by using Qwen2.5-VL for rewards  while testing on separate benchmarks, ensuring our results reflect genuine generative capability rather than metric overfitting.
>
> **Weakness 2: The experiments seem mostly focused on the compositional benchmark, evaluation on broader text-to-image tasks are very limited.**
>
> We did validate our model's performance on a broad, general-purpose text-to-image benchmark.
>
> We kindly refer the reviewer to **Table 3** in the main paper, captioned **"General image quality evaluation on COCO 2014 validation set"**. This experiment evaluates our model on the standard **COCO 2014 validation set** (30k images). This is a diverse, broad-domain benchmark, not a compositional one. On this benchmark, GoT-R1-7B shows significant improvements in general-purpose metrics, including **CLIP Score** (from 29.97 to 31.83) and **Aesthetic Score** (from 5.19 to 5.41). Most importantly, in a **Human Evaluation** on 300 randomly selected prompts, our model was **preferred 77%** of the time over the strong baselines. This demonstrates that our approach not only fixes compositional failures but also enhances general image quality and prompt alignment on broader T2I tasks.
>
> **Question 1: For equation (3), the total reward is a sum ($R_{sem} + R_{spa}$) which is [0,2] times a bunch of other [0,1]s, wouldn't this make the total reward be [0,2]?**
>
> In our actual implementation, the prompt-reasoning reward, $R_{PR}$, is calculated as the **average** of its two components:
>
> $R_{PR} =  \frac{(R_{sem} + R_{spa})}{2}$
>
> This brings the $R_{PR}$ component back into the [0, 1] range, ensuring the final $R_{total}$ also remains within [0, 1]. We have clarified it in the paper.
> A further note is that applying the $\frac{1}{2}$ scaling factor does not affect the optimization of GRPO. It only rescales the numbers, while the relative differences between rewards stay the same. As a result, the policy learns in the same way, and the final performance is unaffected.

---

> > ### Comment · Reviewer_bbst · 2025-11-25
> >
> > Thank you for the detailed rebuttal and explanations. I apologize Weakness 1 as I reviewed the guidelines about concurrent works, All my concerns are addressed, I will raise my score.

---

> > > ### Author Response · Authors · 2025-11-26
> > > **Thank you for acknowledging our rebuttal efforts.**
> > >
> > > Dear Reviewer bbst,
> > >
> > > Thank you very much for updating your rating after the rebuttal. The revised manuscript has been updated on OpenReview. We truly appreciate your thoughtful feedback and are glad that our responses addressed your concerns.
> > >
> > > Best regards,
> > >
> > > The Authors

---

### Official Review · Reviewer_y7UA · 2025-10-26

**Soundness:** 2
**Presentation:** 2
**Contribution:** 2
**Rating:** 4
**Confidence:** 3

**Summary:**

This paper introduces ​​GoT-R1​​, a novel framework that enhances the ability of autoregressive visual generation models to handle complex, compositional text prompts. The key innovation is a ​​dual-stage, multi-dimensional reward function design for GRPO training​​ that uses a Multimodal LLM (MLLM) to evaluate both the reasoning processand the final image.

**Strengths:**

- The paper leverages the proven success of RL (GRPO) in enhancing reasoning for language models like DeepSeek-R1and applies it to the challenging domain of compositional visual generation. The motivation is clear: predefined reasoning templates are a bottleneck, and RL is a natural solution for discovering better strategies.

**Weaknesses:**

- No code and data is provided to reproduce the experiments.
- The approach is computationally expensive. Training involves sampling multiple candidates (N=16) per prompt, each evaluated by a 7B parameter MLLM (Qwen2.5-VL) as the reward model.
- The paper directly applies the original GRPO algorithm with an improved reward function to the new model. As a technical contribution, the effort remains insufficient, with the primary novelty being several rewards.
- The paper mentions T2I-R1 as concurrent work but does not provide a detailed comparison. Since T2I-R1 also uses RL for CoT in image generation, a clearer distinction of GoT-R1's unique contributions.

**Questions:**

- Why the total reward is defined as the product of individual rewards? Further exploration is warranted regarding why addition yields less than multiplication, ideally with more specific case studies on rewards.
- How well does the reward model generalize to prompts that are out-of-domain or contain concepts not well-represented in the MLLM's training data?
- Could you provide some failure cases for GoT-R1? I think analyzing a case where even the RL-trained model fails could reveal valuable insights into the remaining limitations of the approach, such as the fundamental constraints of the base autoregressive architecture or the reward design.
- Can the rewards designed in this paper be used in other GRPO-like methods such DAPO [1]?


[1] DAPO: An Open-Source LLM Reinforcement Learning System at Scale

---

> ### Author Response · Authors · 2025-11-19
> **Rebuttal (Weaknesses)**
>
> We sincerely thank the reviewer for their feedback and insightful questions. We are glad they recognized our clear motivation and the novelty of applying RL to this challenging domain. We appreciate the opportunity to clarify these important points.
>
> ------
>
> **Weakness 1: No code and data is provided to reproduce the experiments.**
>
> We are fully committed to open-sourcing all our contributions for reproducibility. Upon the paper's acceptance, we will release the full source code, pre-trained model checkpoints, and all data required to reproduce our experiments.
>
> ------
>
> **Weakness 2: The approach is computationally expensive. Training involves sampling multiple candidates (N=16) per prompt, each evaluated by a 7B parameter MLLM (Qwen2.5-VL) as the reward model.**
>
> We respectfully disagree and would argue that our framework is, in fact, highly efficient. Our entire GRPO training phase consists of only 1000 steps. This was completed in just 2 days using only 8 L40S-48G GPUs. We believe this is a very modest computational cost compared to the SFT or pre-training phases of most large-scale generative models.
>
> Reinforcement learning is highly step-efficient in this context because it is unlocking the model's inherent reasoning capabilities, not trying to brute-force memorization of a large SFT dataset. Our results show these 1000 steps of GRPO are far more effective than SFT. Moreover, The N=16 sampling is highly parallelizable and fast.
>
> ------
>
> **Weakness 3 & 4: The primary novelty being several rewards... T2I-R1 as concurrent work but does not provide a detailed comparison... a clearer distinction of GoT-R1's unique contributions.**
>
> We thank the reviewer for this critical question, as it allows us to highlight the fundamental methodological differences between our work and the concurrent T2I-R1. Our contribution is not just "several rewards," but a novel, unified, and more robust reward framework.
>
> 1. Our reward framework is built on the insight that supervising the entire generation process (Prompt, Reasoning, Image) is crucial. Our reward is not a simple prompt-image alignment. We explicitly reward the Prompt-Reasoning link and the Reasoning-Image link separately. This is a key technical contribution that, as our ablations show, is essential for success.
> 2. Our approach is a more elegant and unified solution. We demonstrate that a single, powerful MLLM (Qwen2.5VL-7B) can serve as a reliable reward model for all complex aspects of this task (semantic, spatial, and image alignment). This contrasts sharply with T2I-R1, which relies on a complex ensemble of numerous different, specialized models.
>
> 3. More importantly, **T2I-R1 employs VQA models as reward signals, which mirror T2I-CompBench's own evaluation metrics.** This creates a high risk, as the model is effectively optimized directly on the test metric. In contrast, GoT-R1 decouples training from evaluation by using Qwen2.5-VL for rewards  while testing on separate benchmarks, ensuring our results reflect genuine generative capability rather than metric overfitting.

---

> ### Author Response · Authors · 2025-11-19
> **Rebuttal (Questions 1-3)**
>
> **Question 1: Why the total reward is defined as the product of individual rewards? Further exploration is warranted regarding why addition yields less than multiplication...**
>
> We did conduct this experiment. We kindly refer the reviewer to our **Ablation Study on Training Details (Table 6)** in the main paper. The row labeled **"Sum reward"** shows the performance of an alternative model trained by summing the rewards instead of multiplying them.
>
> Our full GoT-R1 model (using multiplication) outperforms the "Sum reward" model across all T2I-CompBench categories, for example, in Color (0.7632 vs. 0.7154), Shape (0.5174 vs. 0.4385), and Spatial (0.2674 vs. 0.2254). This result confirmed that multiplication was a more effective design for our framework.
>
> A possible explanation is that in different experiments, we consistently observe that the **relative improvement ratio** of the rewards remains similar (around 50%). However, the **absolute numerical increase** varies significantly due to different distribution of rewards.
>
> ------
>
> **Question 2: How well does the reward model generalize to prompts..?**
>
> Modern MLLMs such as Qwen2.5-VL exhibit strong generalization beyond their supervised training data. Thanks to large-scale and heterogeneous pre-training, they are known to handle a wide range of prompt styles, concepts, and compositional structures with stable behavior, even when the inputs differ moderately from those seen during pre-training.
>
> Moreover, our RL training does not introduce genuinely out-of-domain prompts. Both T2I-CompBench and LAION-Aesthetics come from the same broad web-scale distribution that Qwen2.5-VL was pretrained on, so the reward model operates strictly within its typical domain. Empirically, we observe stable and consistent reward behavior across all prompt sources used in our experiments. While any pretrained model has limitations on extreme OOD concepts, such cases are not encountered in our setup and do not affect GRPO training.
>
> We have also conducted an experiment to evaluate the reliability of our reward model. We kindly direct the reviewer to the **Appendix, Section B**.
>
> ```
> To validate our reward model choice, we measured correlation with human judgment using Kendall's tau and Spearman's rho on 500 T2I-CompBench prompt-image pairs.
> ```
>
> Furthermore, to validate this choice, we conducted a new experiment during the rebuttal period with another powerful MLLM, InternVL3-8B and our base model Janus-Pro-7B.
>
> | **Model**| **Kendall's tau (τ)** | **Spearman's rho (ρ)** |
> | ------------------- | --------------------- | ---------------------- |
> | Janus-Pro-7B | 0.1451| 0.1547 |
> | CLIP Score| 0.1810| 0.2141|
> | InternVL3-8B| 0.2759| 0.3257 |
> | Qwen2.5VL-7B (Ours) | **0.3147**| **0.3428**|
>
> This new result further strengthens our confidence that Qwen2.5VL-7B is a robust and effective choice for our reward function.
>
> ------
>
> **Question 3: Could you provide some failure cases for GoT-R1?**
>
> We would like to present the some issues of RL training as well as some failure case of our model:
>
> - **Reward Hacking:** Reinforcement learning methods are particularly susceptible to reward hacking. To illustrate this, we conducted an experiment (reported in the **Appendix, Table 9**) where we trained a model using only HPSv2 as the reward signal. While the HPSv2 score did increase, we observed a classic case of reward hacking: the model began generating hyper-saturated and intensely colorful images, a known bias of the HPS metric. This aesthetic preference came at the expense of compositional accuracy (CLIP score actually decreased), confirming that relying on a single metric is insufficient. This experiment is what validated our final multi-dimensional reward design.  It is crucial to note that our evaluation metrics are decoupled from our reward model. The training reward is provided by Qwen2.5VL-7B, while final performance is judged by separate benchmark evaluators (T2I-CompBench, General, Aesthetic Score, Human Evaluation etc.). Therefore, even if the model were to overfit and "hack" the specific reward signals from Qwen2.5VL-7B, this would not guarantee high scores on the distinct evaluation metrics, ensuring a more robust and honest assessment.
> - **Other Failure case of our model:** While GoT-R1 is significantly more robust, it can still struggle with:
>   -  Prompts with a very high number of objects (e.g., >10) can lead to some entities being dropped or merged, as the bounding boxes predicted become unsatisfactory.
>   -  The model can sometimes misinterpret deeply nested or ambiguous spatial descriptions (e.g., "the chair to the immediate left of the farthest window").

---

> > ### Author Response · Authors · 2025-11-19
> > **Rebuttal (Question4)**
> >
> > **Question 4: Can the rewards designed in this paper be used in other GRPO-like methods such DAPO?**
> >
> > Our contribution is the **Dual-Stage Multi-Dimensional MLLM Reward Framework**, which computes a scalar reward for a generated trajectory (Reasoning + Image). This reward signal is mathematically compatible with any RL optimization algorithm that relies on trajectory-level or step-level rewards, including DAPO. DAPO is a GRPO-based variant that retains group-relative advantage estimation while modifying the clipping strategy and removing the explicit KL penalty to improve training stability. Since our reward function simply outputs a value representing the alignment of the generated content, it can directly replace the reward component in the DAPO objective function without modification. We are confident our reward design would provide similar benefits in stabilizing and guiding reasoning-aware generation in a DAPO framework.

---

> > > ### Comment · Reviewer_y7UA · 2025-11-27
> > >
> > > Thank you for your response. I think your response address my questions. I have raised up my score to 6.

---

> > > > ### Author Response · Authors · 2025-11-27
> > > > **Thank you for acknowledging our rebuttal efforts.**
> > > >
> > > > Dear Reviewer y7UA,
> > > >
> > > > Thank you very much for updating your rating after the rebuttal. The revised manuscript has been updated on OpenReview. We truly appreciate your thoughtful feedback and are glad that our responses addressed your concerns.
> > > >
> > > > Best regards,
> > > >
> > > > The Authors

---

### Official Review · Reviewer_hFob · 2025-10-31

**Soundness:** 3
**Presentation:** 3
**Contribution:** 3
**Rating:** 6
**Confidence:** 3

**Summary:**

This paper introduce GOT-R1 a framework apply reinforcement learning to enhance semantic special reasoning in autoregressive visual generation models. This method is build based on the generation chain-of-thought framework, which first generated the reasoning process given the prompt, and then generates images following the reasoning process. The motivation of incorporating reinforcement learning into the reasoning change generation is that the author observe that GoT-generated reasoning chains can be unfaithful to the prom, despite following templates well. Thus authors introduce GoT-R1, adapting reinforcement, learning, advances to enhance semantic spatial reasoning in autoregressive vision generation. Specifically, in GoT-R1, there is a comprehensive reward framework to ensure that effective of training, including rewards for prompt to reasoning semantic alignment, prompt-reasoning spatial alignment, reasoning–to–image alignment, and prompt-to-image alignment. Experiment results demonstrate that GoT-R1 achieves state of the art on T2I-CompBench and Geneval.

**Strengths:**

1. This paper introduces an effective way to enhance the chain-of-thought reasoning capability of autoregressive visual generation model. Experiment results directly support the effectiveness of the proposed method.
2. The reward framework is comprehensive, cross alignment between prompt, reasoning chain and images are considered. This rewards are general, and could provide inspiration to the rewards design of other visual generation model.
3. The paper is well organised, clearly written and easy to follow and understand.

**Weaknesses:**

1. The experiments are conducted with Janus-Pro-1B and Janus-Pro-7B as the base model. It'll be interesting to see how the GoT-1 framework would perform with other base models. (I know it is not a necessary experiment.)
2. The experiments really demonstrates that the GoT-R1 method help improve the quality of the final generated image. While the quality of chain of thought is not evaluated, especially compared to the original GoT.
3. Some of reference of the table in the main text are wrong, for example,  table 4.4 in line 433, table 4.5 and line 445.

**Questions:**

1. In line 322, it is said that the parameter of MLLM are updated through LoRA finetune. As I understand, the NLLM is used as the reward model which should be fixed. Why the MLLM should be finetuned?
2. During the experiment, Qwen2.5VL-7B is selected as the reward model. Have you evaluate how reliable Qwen2.5VL-7B is as the reward model?

---

> ### Author Response · Authors · 2025-11-19
> **Rebuttal response to Reviewer hFob**
>
> We sincerely thank the reviewer for their positive feedback and insightful questions. We appreciate this opportunity to provide clarifications on the excellent points raised.
>
> ------
>
> **Weakness 1: The experiments are conducted with Janus-Pro... It'll be interesting to see how the GoT-R1 framework would perform with other base models.**
>
> The reviewer raises a great point about the generalization of our framework. Our GoT-R1 framework has specific architectural requirements that guided our choice:
>
> 1. The model must be a unified model capable of predicting interleaved text and image.
> 2. The model's image generation process should be auto-regressive token generation. This architecture is necessary because the reinforcement learning loss must be backpropagated through the entire generation process (both text and image tokens). This is hard for standard diffusion models or rectified flow models.
>
> At the time of our submission, Janus-Pro was the state-of-the-art model that met these specific criteria. We are fully confident that our GoT-R1 framework is a general method that can be successfully applied to any future autoregressive, unified text-image models that share this architecture.
>
> ------
>
> **Weakness 2: ...the quality of chain of thought is not evaluated, especially compared to the original GoT.**
>
> We thank the reviewer for this question, as it allows us to highlight a key evaluation that directly addresses this point. We did, in fact, include a head-to-head comparison of the reasoning quality in the main paper.
>
> We kindly direct the reviewer to **Section 4.4: "ANALYSIS ON SELF-EXPLORED GENERATION CHAIN-OF-THOUGHT"**. In this section, we used **GPT-4o** to evaluate the quality of the GoT (reasoning) content generated by our RL-trained **GoT-R1-7B** versus the SFT-baseline **Janus-Pro-7B-GoT**. As detailed in **Table 5**, the results show that our model's self-explored reasoning is overwhelmingly preferred by GPT-4o across all evaluated categories.  This quantitatively confirms that our RL process leads to a significant improvement in the quality of the reasoning chain itself.
>
> ------
>
> **Weakness 3: Some of reference of the table in the main text are wrong...**
>
> We sincerely thank the reviewer for catching these typographical errors. We have already fixed them in the paper.
>
> ------
>
> **Question 1: In line 322, it is said that the parameter of MLLM are updated through LoRA... the MLLM is used as the reward model which should be fixed. Why the MLLM should be finetuned?**
>
> The "MLLM" being updated with LoRA in Line 322 refers to our **base generation model**, which is **Janus-Pro-7B**. This model is a unified MLLM, and we use LoRA for efficient training of its parameters during RL.
>
> The reward model Qwen2.5VL-7B is a separate MLLM and is kept **completely frozen** during the RL process. We have revised the text in the paper.
>
> ------
>
> **Question 2: During the experiment, Qwen2.5VL-7B is selected as the reward model. Have you evaluate how reliable Qwen2.5VL-7B is as the reward model?**
>
> We did conduct this analysis, as it is vital for validating our results. We kindly direct the reviewer to the **Appendix, Section B**, titled **"MLLM Reward Model Reliability"** in the appendix.
>
> As shown in **Table 7 of the Appendix**, we measured the correlation of our reward model with human judgments on 500 T2I-CompBench prompt-image pairs.
>
> ```
> To validate our reward model choice, we measured correlation with human judgment using Kendall's tau and Spearman's rho on 500 T2I-CompBench prompt-image pairs. Table 1 shows our Qwen2.5VL-7B reward model achieves higher alignment with human preference compared to CLIP Score.
> ```
>
> Furthermore, to validate this choice, we conducted a new experiment during the rebuttal period with another powerful MLLM, InternVL3-8B and our base model Janus-Pro-7B.
>
> | **Model**           | **Kendall's tau (τ)** | **Spearman's rho (ρ)** |
> | ------------------- | --------------------- | ---------------------- |
> | Janus-Pro-7B        | 0.1451                | 0.1547                 |
> | CLIP Score          | 0.1810                | 0.2141                 |
> | InternVL3-8B        | 0.2759                | 0.3257                 |
> | Qwen2.5VL-7B (Ours) | **0.3147**            | **0.3428**             |
>
> This new result further strengthens our confidence that Qwen2.5VL-7B is a robust and effective choice for our reward function. We hope these clarifications fully address the reviewer's questions, and we thank them again for their constructive feedback.

---

### Official Review · Reviewer_Xzzf · 2025-11-01

**Soundness:** 3
**Presentation:** 3
**Contribution:** 3
**Rating:** 6
**Confidence:** 4

**Summary:**

This paper proposes a reinforcement learning framework for training Generation Chain-of-Thought (GoT) capabilities in image generation. The base model used in this framework is a unified generative understanding model, which leverages the base model's ability to generate text, coordinates, and images. A dual-stage multi-dimensional reward framework is designed and trained using the GRPO algorithm. Experimental results show that the proposed framework delivers significant performance improvements on T2I-CompBench and GenEval.

**Strengths:**

1. This paper uses reinforcement learning to enhance Generation Chain-of-Thought (GoT) capabilities, which is an interesting and meaningful perspective in the field of multimodal generation.
2. This paper proposes a dual-stage multi-dimensional reward framework that comprehensively utilizes the prompts, reasoning content, and images in spatial and semantic alignment. This design effectively utilizes the capabilities of the unified generative understanding model.

**Weaknesses:**

1. The paper does not specify the number of GRPO training steps or the amount of data used. No curves illustrating the GRPO training process are shown.

2. The authors used a model that unifies generative understanding, but did not evaluate the performance of the trained model in multimodal understanding. Why is that? I am very interested in how the multimodal understanding capability changes after training with reinforcement learning on GOT.

3. The ablation experiments shown in Table 4 are puzzling. For example, row 5 (w $R_{PI}$) plays a crucial role, but row 8 (w $R_{RI}$ & $R_{PI}$) with the addition of $R_{RI}$ performs worse, and the penultimate row further weakens the gain. However, adding $R_{sem}$ at the end reverses this and becomes the best performance. The ablation experiment results here are strange, making me doubt the reliability of the experiment.

**Questions:**

1. Refer to the issues raised in the weakness section.

2. Why is multiplication used between rewards in Equation 3? Summation is generally used now. Experiments are needed to verify this. The curves showing the changes in rewards during training should also be added.

3. Why not utilize the generative understanding of the unified model's own understanding capabilities to obtain rewards, thus avoiding reliance on an additional large multimodal model as the reward model?

---

> ### Author Response · Authors · 2025-11-19
> **Rebuttal(Weaknesses)**
>
> We sincerely thank the reviewer for their valuable feedback. We appreciate the opportunity to clarify the excellent questions raised regarding our training, ablations, and design choices.
>
> ------
>
> **Weakness 1: The paper does not specify the number of GRPO training steps ....**
>
> We would like to clarify that some of the details are included in the **Training Settings (Section 4.1)**. Specifically,
>
> 1. Our model is trained on 70000 steps of SFT and 1000 steps of GRPO.
> 2. The SFT dataset consists of data from LAHR-GoT, JourneyDB-GoT and FLUX-GoT. The prompts for GPRO is from T2I-Compbench training dataset (4200 prompts) and Laion Aesthetic prompts (We sampled 1000 prompts for general image generation purpose).
>
> We agree that a training curve is valuable. **We have already added the reward curve to the Appendix (Figure 7).** To facilitate clearer comparison across training progress, we rescale the reward values such that the reward at step 0 is mapped to 0, and the reward at step 1000 is mapped to 1.
>
> | Step | Rescaled Total Reward (rescaling step 0 to 0 and step 1000 to 1) |
> | ---- | ------- |
> | 0    | 0.000 |
> | 200  | 0.4850|
> | 400  | 0.7700|
> | 600  | 0.9540|
> | 800  | 0.9605|
> | 1000 | 1.0000|
>
> This presents a smooth increase of the reward as the step count increases, indicating stable performance gains over longer trajectories.
>
> ------
>
> **Weakness 2: The authors used a model that unifies generative understanding, but did not evaluate the performance of the trained model in multimodal understanding** ...
>
> This is a very insightful question. The GoT-R1 model undergoes a two-stage training process: an initial SFT stage on GoT data, followed by a RL stage on T2I-Compbench and LAION-Aesthetic prompts. Because our work primarily focuses on enhancing the model’s generative ability, the SFT stage is designed to specialize the model for generation. However, this specialization comes at a cost: the model’s general multimodal understanding capability becomes weaker than that of the original Janus-Pro model.
>
> To address this concern, we further extend our study by incorporating multimodal understanding data directly into the GoT-RL pipeline. Specifically, we construct GoT-R1-7B-Uni by adding samples from the General category of Open-Bee/Honey-Data-15M, including Co-Instruct, LLaVA-Instruct-300k, LLAVA-NeXT-Data, GQA, IconQA, EST-VQA, Objects365, LVIS-Instruct, and ShareGPT4o/ShareGPT4V-Knowledge. During the SFT stage, we mix generation and understanding tasks in an 8:2 ratio over 100k steps. In the subsequent RL stage, we include an additional 20% of SFT-style understanding steps to further reinforce the model’s comprehension ability.
>
> We then evaluate GoT-R1-7B-Uni on standard multimodal understanding benchmarks as well as T2I-Compbench. Thanks to the strong regularization on visual understanding from Honey-15M. The results show that its understanding capability remains comparable to Janus-Pro-7B, while still preserving the substantial improvements in image generation.
>
> |               | POPE | MMBench | MMMU | Color  | Shape  | Texture | Spatial | Non-Spatial | Complex |
> | ------------- | ---- | ------- | ---- | ------ | ------ | ------- | ------- | ----------- | ------- |
> | Janus-Pro-7B  | 87.4 | 79.2    | 41.0 | 0.6359 | 0.3528 | 0.4936  | 0.2061  | 0.3085      | 0.3559  |
> | GoT-R1-7B-Uni | 85.1 | 76.7    | 43.2 | 0.7645 | 0.5013 | 0.6842  | 0.2874  | 0.3107      | 0.3788  |
>
> ------
>
> **Weakness 3: The ablation experiments shown in Table 4 are puzzling... the ablation experiment results here are strange, making me doubt the reliability of the experiment.**
>
> We appreciate the reviewer's close reading of the ablation study.  We argue these "strange" results are not a sign of unreliability, rather, they directly validate our core motivation for this paper.
>
> 1. Training only with the Prompt-Image reward in row 5 provides a decent boost. This is expected, as it's the most direct, end-to-end signal.
> 2. The most critical finding is row 8. Here, performance drops compared to using prompt-image reward alone. This is our key motivation.  As we state and show in Figure 1, the SFT-baseline reasoning chains are often unfaithful to the prompt. This reward setup forces the model to align its image with a flawed reasoning plan, which is detrimental.
> 3. This is exactly why the Prompt-Reasoning reward is essential. Adding semantic and spatial rewards starts to fix the reasoning plan itself, ensuring the plan is semantically/spatially correct.
>
> The final row (our full GoT-R1) adds all components. It first ensures the reasoning plan is correct  and then ensures the image aligns with that corrected plan and original prompt. This step-by-step logic demonstrates that our full, multi-stage reward is necessary to first fix the reasoning and then execute it, proving the reliability and necessity of our complete framework.

---

> ### Author Response · Authors · 2025-11-19
> **Rebuttal (Questions)**
>
> **Question 1: Why is multiplication used between rewards in Equation 3? Summation is generally used now. Experiments are needed to verify this.**
>
> We thank the reviewer for this question, as we did test this. We kindly refer to the **Ablation Study on Training Details (Table 6)**. The row labeled **"Sum reward"** shows the performance of an alternative model trained by summing the rewards instead of multiplying them. Our full model ("GoT-R1") outperforms the "Sum reward" model across the board, for example, in Color (0.7632 vs 0.7154), Shape (0.5174 vs 0.4385), and Spatial (0.2674 vs 0.2254). This experiment confirms that using the product was a more effective design for our framework.
>
> ------
>
> **Question 2: Why not utilize the generative understanding of the unified model's own understanding capabilities to obtain rewards...?**
>
> This is a good question. We did attempt this during our initial experiments. Unfortunately, we found that the Janus-Pro-7B model's own understanding capabilities, when used as a reward function for this specific task, were not reliable. It tended to produce "random scores" that were not well-correlated with actual compositional quality, which hindered or even destabilized the RL training. We found that Qwen2.5VL-7B was significantly more powerful and reliable as an evaluator. As we show in the **Appendix (Table 7)**, Qwen2.5VL-7B has a much stronger correlation with human judgment than standard metrics, making it a necessary choice for effective training.
>
> We would like to also present the prompt-image scoring **correlation with human judgements** of Janus-Pro-7B and InternVL3-8B compared with Qwen2.5VL-7B to justify our reward model selection. All the experiments are conducted on 500 text-image pairs as detailed in Appendix Section B.
>
> | **Model**           | **Kendall's tau (τ)** | **Spearman's rho (ρ)** |
> | ------------------- | --------------------- | ---------------------- |
> | Janus-Pro-7B        | 0.1451                | 0.1547                 |
> | CLIP Score          | 0.1810                | 0.2141                 |
> | InternVL3-8B        | 0.2759                | 0.3257                 |
> | Qwen2.5VL-7B (Ours) | **0.3147**            | **0.3428**             |
>
> We further compare the rewards assigned by Janus-Pro-7B with those produced by our GoT-R1-7B. Both models are trained for 1000 GRPO steps, initialized from the same SFT checkpoint, Janus-Pro-7B-GoT.
>
> | T2I-Compbench Eval           | Color      | Shape      | Texture    | Spatial    | Non-Spatial | Complex    |
> | ---------------------------- | ---------- | ---------- | ---------- | ---------- | ----------- | ---------- |
> | Janus-Pro-7B-GoT             | 0.6551     | 0.5008     | 0.5836     | 0.2457     | 0.3113      | 0.3754     |
> | Janus-Pro-7B as reward model | 0.6510     | 0.5195     | 0.5671     | 0.2271     | 0.3124      | 0.3740     |
> | GoT-R1-7B                    | **0.8139** | **0.5549** | **0.7339** | **0.3306** | **0.3169**  | **0.3944** |
>
> The results show that leveraging an inaccurate reward model may even lead to degradation in performance which justifies our choice.

---

### Author Response · Authors · 2025-12-03
**Review and Reviewer-Author Discussion Summary**

Dear PCs, SACs, ACs, and Reviewers,

Thank you very much for your valuable contributions to our work. To assist the newly assigned AC and help reduce their workload, we provide below a summary of the key points from the reviews and the reviewer-author discussions.

**Strengths.** Reviewers expressed positive feedback and collectively recognized the following key strengths of our work:

1. **Novel and Effective Method:** Reviewers commended the paper for introducing an "interesting and meaningful perspective" and an "effective way to enhance the chain-of-thought reasoning capability" of autoregressive models through reinforcement learning. The also praised that our motivation is clear.

2. **Comprehensive Reward Design:** The reviewers praised the "dual-stage multi-dimensional reward framework", noting that it is "comprehensive" and could "provide inspiration to the rewards design of other visual generation models".

3. **Significant Performance & Clear Validation:** Reviewers confirmed that our method shows "significant improvement on compositionality" and that the results "directly support the effectiveness of the proposed method". The ablation studies were also recognized as "clear" and supportive of the design choices.

**Key Concerns and Our Addressing.** Building on these strengths, our rebuttal addressed the remaining questions regarding novelty, methodology, and robustness, leading to confirmed score increases:

1. **Novelty relative to concurrent works (Reviewers y7UA W4 & bbst W1)** We clarified the fundamental distinction between GoT-R1 and concurrent works (e.g., T2I-R1). While concurrent methods often rely on outcome-only supervision (prompt–image reward) or a complex ensemble of various vision experts (including a VQA model optimized directly on the test metric), **GoT-R1 implements comprehensive process supervision via a single, unified MLLM**. We demonstrated that supervising the intermediate reasoning chain is critical for stable generation.

2. **Justification of reward aggregation (Reviewers Xzzf Q2 & y7UA Q1)** We provided empirical evidence validating our mathematical design. Ablation studies (Table 6) demonstrated that our multiplicative reward formulation consistently outperforms summation, proving that a coupled reward structure effectively enforces simultaneous semantic and spatial alignment.

3. **Robustness and Efficiency (Reviewers Xzzf Q3; hFob W2, Q2 & y7UA W2 Q4)** We addressed concerns regarding training costs and reward reliability. We clarified that our RL training is highly efficient (less than 1000 steps). Furthermore, we included more models for comparison in Appendix B, showing that our reward model Qwen2.5-VL achieves significantly higher correlation with human judgment than standard metrics like CLIP and other SOTA MLLMs. We also highlighted improvements on the COCO validation dataset and reasoning-chain quality to demonstrate strong generalization. Last but not least, we explained that our method is capable of transferring to other RL methods like DAPO.

We have also included the training curves (Reviewer Xzzf W1), corrected typographical errors (Reviewer hFob W3), and clarified the relevant details (Reviewer hFob Q1 & Reviewer bbst Q1) in the revised manuscript. We further provided detailed responses to all remaining questions and points of confusion raised by the reviewers.

**Recognition of our revision from reviewers.**  Following these clarifications and additional experiments, Reviewer y7UA raised their score from 4 to 6, and Reviewer bbst increased their score from 2 to 6. Reviewer Xzzf and hFob have not updated their reviews but already assigned positive score of 6 in their original assessments.

We sincerely thank you again for the time and effort, which has significantly strengthened our paper.

Sincerely,

The Authors

---

### Meta-Review · Area_Chair_b4Z1 · 2026-01-06

**Summary:**

In the initial stages, this paper received mixed reviews (6 / 6 / 4 / 2). The reviewers noted the performance gains shown by the method and praise the general approach of applying RL to improve the semantic reasoning of AR models. Most of the concern appear relatively direct and are directly addressed in the rebuttal (e.g. not specifying the # of GRPO training steps, asking about ablations that are already present in the paper). The main concerns listed are the generalization to other base models and the novelty of the technique.

**Reviewer Concerns:**

- For the base model, the authors note that their method require a unified model capable of predicting interleaved text and image which is autoregressive.
- For the novelty, the main comparisons used by the reviewer were concurrent work, which were not accepted yet at the time of submission.

**Reviewer Scores:**

Several reviewers noted that their scores would increase. Generally it seems like the concerns listed by the reviewers were addressed.

---

### Decision · Program_Chairs · 2026-01-26

Accept (Poster)